# A Story of Kinases and Adaptors: The Role of Lck, ZAP-70 and LAT in Switch Panel Governing T-Cell Development and Activation

**DOI:** 10.3390/biology12091163

**Published:** 2023-08-24

**Authors:** Luis M. Fernández-Aguilar, Inmaculada Vico-Barranco, Mikel M. Arbulo-Echevarria, Enrique Aguado

**Affiliations:** 1Institute for Biomedical Research of Cadiz (INIBICA), 11009 Cadiz, Spain; luismi.fernandez@uca.es (L.M.F.-A.); inmaculada.vicobarranco@gmail.com (I.V.-B.); mmartinezdeae@outlook.com (M.M.A.-E.); 2Department of Biomedicine, Biotechnology and Public Health (Immunology), University of Cadiz, 11002 Cadiz, Spain

**Keywords:** TCR, CD3, ITAMs, Lck, ZAP70, LAT, signaling

## Abstract

**Simple Summary:**

Tyrosine phosphorylation is the first biochemical event that occurs after TCR engagement, which is crucial for T-cell development, activation and differentiation. Early TCR signals include phosphorylation events in which the tyrosine kinases Lck and ZAP70 are involved. The sequential activation of these kinases leads to the phosphorylation of the transmembrane adaptor LAT, which constitutes a signaling hub for the generation of a signalosome, finally resulting in T-cell activation. The negative regulation of these early signals is key to avoid aberrant processes that could generate inappropriate cellular responses and disease. In this review, we examine and discuss the roles of the tyrosine kinases Lck and ZAP70 and the membrane adaptor LAT in the TCR-signaling cassette, both of their functions in activation signal transduction and the negative-feedback loops in which they participate. A better knowledge of these negative regulatory mechanisms may be critical not only for understanding T-cell activation, but also for a more efficient design of therapeutic approaches and a better understanding of some immune-based pathologies.

**Abstract:**

Specific antigen recognition is one of the immune system’s features that allows it to mount intense yet controlled responses to an infinity of potential threats. T cells play a relevant role in the host defense and the clearance of pathogens by means of the specific recognition of peptide antigens presented by antigen-presenting cells (APCs), and, to do so, they are equipped with a clonally distributed antigen receptor called the T-cell receptor (TCR). Upon the specific engagement of the TCR, multiple intracellular signals are triggered, which lead to the activation, proliferation and differentiation of T lymphocytes into effector cells. In addition, this signaling cascade also operates during T-cell development, allowing for the generation of cells that can be helpful in the defense against threats, as well as preventing the generation of autoreactive cells. Early TCR signals include phosphorylation events in which the tyrosine kinases Lck and ZAP70 are involved. The sequential activation of these kinases leads to the phosphorylation of the transmembrane adaptor LAT, which constitutes a signaling hub for the generation of a signalosome, finally resulting in T-cell activation. These early signals play a relevant role in triggering the development, activation, proliferation and apoptosis of T cells, and the negative regulation of these signals is key to avoid aberrant processes that could generate inappropriate cellular responses and disease. In this review, we will examine and discuss the roles of the tyrosine kinases Lck and ZAP70 and the membrane adaptor LAT in these cellular processes.

## 1. Introduction

T lymphocytes are major components of immune responses via their ability to specifically recognize foreign, potentially dangerous antigens. In order to perform such specific detection, T cells express at their surface an antigen receptor called the T-cell receptor (TCR) [1,2]. The TCR expressed at the plasma membrane in the majority of T cells is composed of a TCR-α and TCR-β heterodimer, which is able to recognize an antigen peptide presented in the binding groove of an MHC molecule. TCRs are clonally distributed, and they are generated during thymic development through site-specific DNA recombination. In this way, each mature T lymphocyte has a unique antigen receptor, different from the others in the rest of T cells [2]. Therefore, the immune system has an immense array of T cells expressing TCRs of different specificities able to recognize and respond to any possible antigen. In order to do so, upon the specific engagement via a peptide–MHC (pMHC) complex, the TCR triggers multiple intracellular signals leading to the activation, proliferation and differentiation of T lymphocytes into effector cells, and the complexity of these signals, together with cytokine signaling, are responsible of the differential fates that T lymphocytes undergo.

Regarding the structure of the TCR, the majority of T cells express a TCRαβ heterodimer for the specific recognition of peptide antigens coupled to MHC molecules. There are also T cells expressing γδ TCRs, which are a minor fraction of T lymphocytes in peripheral blood, although they are present at an increased frequency in mucosal tissues and the skin and have immune functions different from those of αβ T cells [3,4]. TCR-α and -β polypeptides belong to the immunoglobulin superfamily, and each contains an amino-terminal variable domain and a carboxy-terminal constant domain, followed by a transmembrane segment and a very short cytoplasmic tail. In order to transduce intracellular signals after pMHC engagement, the constant domains of TCR-α and TCR-β associate with CD3ε, CD3γ, CD3δ and ζ-chain polypeptides, organized as dimers: CD3δε, CD3γε and ζ-ζ homodimers [5]. All these polypeptides contain in their transmembrane segments charged amino acids, either negatively charged (in TCR-α and TCR-β) or acidic residues (in CD3 polypeptides or the ζ-chain), allowing for the non-covalent interaction of TCRαβ with CD3 proteins, and constituting the TCR/CD3 complex. Given that α and β polypeptides have short cytoplasmic tails lacking the ability to process this information to the cells, interaction with CD3 dimers is essential to transduce these signals coming from the TCR, as their cytoplasmic tails contain one or several immunoreceptor tyrosine-based activation motifs (ITAMs) [6] (Figure 1). ITAMs are formed by conserved sequences of amino acids (Yxx(L/I)x6-8Yxx(L/I)), with two tyrosine residues, which become phosphorylated after TCR engagement, and this constitutes one of the earliest signaling events that occur after the activation of T cells.

ITAMs constitute signaling modules coupled to other types of immune receptors, such as the Fc-γR, Igα, Igβ or DAP12 [7]. After the binding of immunoreceptors to their ligands, ITAMs are phosphorylated by the Src family of protein tyrosine kinases (SFKs). The SFK family contains several members expressed in different hematopoietic lineages and includes Lyn, Fyn, Lck, Blk, Hck, Src, Fgr and Yes [8]. T cells primarily express both Lck and Fyn, but whereas Fyn colocalizes in the cytoplasm with centrosomal and mitotic structures, Lck is mainly located at the plasma membrane and is currently accepted as the main contributor to ITAMs phosphorylation in the TCR/CD3 complex [8,9] (Figure 2A). ITAMs phosphorylation generates binding motifs for the SH2 domains of the Syk family of Tyr kinases, which includes Syk and ZAP70 (Figure 2B). T cells express ZAP70, and, upon recruitment to phosphorylated ITAMs, this kinase is phosphorylated and activated [8]. Subsequently, ZAP70 phosphorylates the membrane adaptor LAT (Linker for the activation of T cells), leading to the formation of the LAT signalosome, including the SLP-76 cytosolic adaptor and phospholipase C-γ1 (PLC-γ1), finally resulting in T-cell activation [8,10] (Figure 2C). Although these activatory signaling pathways have been intensely examined, many questions remain to be elucidated with regard to the regulatory mechanisms, which prevent excessive immune responses, potentially leading to uncontrolled and potentially harmful immune responses. Here, we review confirmed data on early TCR signaling and several regulatory mechanisms involving Lck and LAT, and also recent findings about the relationship between both molecules.

## 2. Initiation of Intracellular Signaling Associated with the TCR/CD3 Complex

As already mentioned, the specific recognition of antigens gives higher vertebrates a way by which individual threats can be eliminated in a cost-effective manner [11]. Specific identification is performed by receptors at the surface of T and B lymphocytes in a coordinated way with other immune elements, allowing the best-suited immune responses for every type of danger. This type of response involves the generation of a vast number of possible specific receptors, which are clonally distributed, and the resulting repertoires are large enough to recognize any infecting microorganism. These receptors, immunoglobulins for B cells and TCRs for T lymphocytes, are expressed on the plasma membrane and are randomly generated by means of the somatic recombination of variable genetic segments.

The stoichiometry of the TCR/CD3 complex is a crucial parameter because it determines the receptor valency (i.e., the number of pMHC-binding units per TCR/CD3 complex). There has long been a great deal of controversy about the stoichiometry of the TCR, although the most accepted is a monovalent model in which there is one TCRαβ heterodimer associated with individual CD3γε, CD3δε and ζ-ζ dimers [4]. Other studies using the measurement of sedimentation coefficients, native gel electrophoresis and immunological analysis of detergent-solubilized receptors, as well as the observation of higher-order TCR oligomers via electron microscopy, suggested that the TCR/CD3 complex may have a bivalent or multivalent nature (reviewed in [5]). New data obtained via different imaging techniques, including single-molecule brightness, single-molecule coincidence analysis and photon-antibunching-based fluorescence correlation spectroscopy, support the monovalent nature of the TCR/CD3 complex and a 1:1:1:1 stoichiometry (TCR-αβ:CD3γε:CD3δε:CD3ζζ) [12]. Additional support for the monovalent model has been provided by means of cryogenic electron microscopy (cryo-EM). Cryo-EM is an electron microscopy technique applied to samples cooled to cryogenic temperatures, which allows the structures of biomolecules to be determined at near-atomic resolution [13]. Using this technology, it has been possible to decipher the structure of the TCRαβ/CD3 complex, confirming that it assembles with a 1:1:1:1:1 stoichiometry [14].

Interestingly, while the structures of the TCRαβ and TCRγδ are very similar, it has been shown that the TCRγδ lacks CD3δ and, consequently, the stoichiometry is different from the TCRαβ [15,16]. This differential stoichiometry could partially explain why naive γδ T lymphocytes are able to react quickly after pathogen contact, playing, in this way, a fast, innate, immunity-like role before αβ T cells and other adaptive immune responses [17]. Although some differences have been reported in the intracellular signaling between αβ and γδ TCRs, we will focus in this review on the former because of the greater knowledge about the molecular mechanisms governing it.

Regarding the initiation of TCR signaling, it was initially thought that Nck was involved in the initiation of TCR signaling through a conformational change [18]; however, there is now ample evidence that this is not the case [19,20]. The same group reported additional data supporting the view that a conformational change occurs in the TCR/CD3 complex after the binding of pMHC to the TCR heterodimer, affecting not only mature T lymphocytes, but also thymocytes [18,21]. Interestingly, it seems that during thymic development, negative selection induces the conformational change in CD3ε, but it is elicited in only a small percentage of immature thymocytes during positive selection, raising the possibility that the TCR/CD3 conformational change could underlie the discrimination of ligands leading to positive and negative selection [22]. The same group previously showed that the conformational change in CD3ε exposes an intracellular epitope, which can be detected with a specific monoclonal antibody (mAb) (APA1/1) that binds to the cytoplasmic tail of CD3ε [18]. Remarkably, the exposure of the APA1/1 epitope is very fast, independent of tyrosine kinase activity, and happens in a restricted area of the immune synapse, strongly supporting a role of this conformational trigger of subsequent intracellular-signaling events [23].

However, beyond the controversy about the conformational change in the TCR/CD3 complex [24], it is also unknown how such putative structural modifications in the TCR/CD3 complex trigger intracellular signaling. It was proposed that CD3 ITAMs are buried into the plasma membrane because of the electrostatic attraction between basic CD3ε residues and acidic phospholipids enriched in the inner leaflet of the plasma membrane [25]. Therefore, the conformational change in the TCR/CD3 complex would result in the detachment of the cytosolic domain of CD3ε from the plasma membrane, allowing for the phosphorylation of tyrosine residues in the ITAM by Lck. Nevertheless, subsequent studies have shown that the sequestration of ITAMs is not involved in the initiation of TCR signaling, but in a positive-regulatory-feedback loop that seems to amplify CD3 phosphorylation and enhance the T-cell sensitivity to foreign antigens [26,27].

In this sense, more support for conformational changes in the TCR/CD3 complex comes from the fact that the juxtamembrane regions of the ζ-ζ homodimer are separated within the TCR/CD3 complex until TCR activation triggers their juxtaposition, transferring information across the cell membrane [28]. More recently, Lanz et al. showed that mutations in the transmembrane regions of TCRβ and the ζ-chain loosen the quaternary structure of the TCR/CD3-complex cohesion and trigger intracellular signaling [29]. In their work, Acuto and colleagues show that these gain-of-function mutations loosen the interaction between TCRαβ and the ζ-chain, and, strikingly, pMHC binding reduces the TCRαβ cohesion with ζ. Recently, Susac et al. used cryo-EM to examine the structure of a TCRαβ/CD3 expressed in Chinese hamster ovary cells bound to a soluble pMHC [30]. This work showed that the antigen-bound TCR/CD3 complex comprises 11 subunits stabilized via multivalent interactions, with clustered membrane-proximal disulfide bonds between cysteine residues stabilizing the CD3δε and CD3γε heterodimers. Interestingly, the comparison of the unliganded and ligand-bound TCR/CD3 structures indicates that, despite some minor conformational rearrangements located at the complementarity-determining regions (CDRs), the TCR is largely unchanged after pMHC binding. The question of conformational changes in the TCR as a trigger of signaling remains an unresolved issue. Cryo-EM should shed more light on this when the structures of TCRs bound to membrane-tethered ligands are eventually solved.

## 3. Lck Goes into Action: The First Biochemical Events of the TCR-Signaling Cascade

There is a broad consensus that the first biochemical changes occurring after the specific recognition of an antigen by a TCR are the phosphorylation of the tyrosine residues included in the ITAM sequences of CD3 and the ζ-chain [1,31,32]. As already mentioned, ITAMs constitute signaling modules containing two tyrosine residues apiece, and the tyrosine kinase Lck is responsible for the phosphorylation of these tyrosines upon TCR engagement. Lck belongs to the Src family of non-receptor tyrosine kinases, also including Src, Blk, Fgr, Fyn, Hck, Lyn, Yes and Yrk. The *Lck* gene was originally cloned in a murine T-cell lymphoma, and it was initially speculated that alterations in the structure or expression of this gene might mediate neoplastic transformation [33]. Indeed, it was very soon revealed that the *Lck* gene is located in human and mice near sites of frequent structural abnormalities in human lymphomas and neuroblastomas [34]. Regarding its biological functions, the relevance of Lck kinase for thymic development became clear very soon, as *Lck* deficiency generated a dramatic reduction in thymic cellularity associated with thymic atrophy, with a significant decrease in the CD4+CD8+ Double Positive (DP), and a decrease in CD4+ and CD8+ T cells in peripheral lymphoid organs [35]. However, T lymphocytes from *Lck*-knockout (*Lck*^−/−^) mice were still able to proliferate in response to anti-CD3 stimulation, although their capacity was decreased with regard to T lymphocytes from wild-type mice. These results showed that Lck was not entirely essential during thymic development, and suggested that another kinase could replace it, at least partially, until the DP stage. Indeed, it seems that Fyn kinase can promote the development of some αβ T cells in *Lck*^−/−^ mice, as the double disruption of the Lck and Fyn genes (*Lck*^−/−^*Fyn*^−/−^) completely blocked thymic maturation at the CD4−CD8− Double-Negative (DN) stage.

If we take a look at its structure, Lck has the typical backbone found in the Src family, with an N-terminal membrane-localization SH4 domain, followed by SH3, SH2 and catalytic domains, and a short carboxy-terminal tail (Figure 3A) [36]. Lck is myristoylated at glycine residue in position 2, and palmitoylated at cysteines 3 and 5, with palmitoylation being essential for membrane attachment and biological activity [37,38]. Cysteine residues in the SH4 domain coordinate a Zn^2+^-ion-dependent CD4/CD8 binding mechanism, which also involves a motif with two cysteine residues present in CD4 and CD8α molecules [39]. Thus, Lck binding to CD4 or CD8 co-receptors suggests a signaling-initiation mechanism following antigenic recognition by the TCR. According to this model, CD4/CD8 binding to non-polymorphic regions on MHC molecules presenting antigenic peptides brings the tyrosine residues in the ITAM motifs into close proximity.

The Lck SH2 domain binds to phospho-tyrosine motifs, and this domain is followed by a linker region and the kinase domain. The kinase domain of Lck contains a tyrosine residue in position 394 (Y394), which can be autophosphorylated, a modification that promotes the activation of kinase activity. Further downstream of the kinase domain is a tyrosine residue essential for the negative regulation of Lck, tyrosine 505 (Y505). Lck activity is mainly controlled via the phosphorylation of the tyrosine residues 394 and 505 (Y394, Y505), and also via conformational changes involving the binding of its SH2 and SH3 domains [40]. Lck adopts an “open” or “closed” conformation that depends on Y505 phosphorylation. The dephosphorylation of Y505 residue allows Lck to unfold, keeping the kinase in a primed conformation, leaving accessible the Y394 residue that requires autophosphorylation for full kinase activity. In this way, Lck acquires an “open” conformation and therefore an active form (Figure 3B). The phosphorylation of Y505 allows for the binding of this tyrosine to the SH2 domain, and this conformation is further stabilized via the interaction of the SH3 domain with a proline motif (PXXP) located in the linker region. Lck thus adopts a “closed” conformation, an inactive form, as the tyrosine 394 of the kinase domain is inaccessible and cannot be phosphorylated for activation [41] (Figure 3C).

The transitions between the different phosphorylation states of Lck are regulated via the action of kinases and phosphatases, as well as via the binding of different molecules to the SH2 and SH3 domains in Lck, establishing a dynamic equilibrium between the different forms of Lck. The kinase Csk is able to phosphorylate Y505 residue, promoting the closed conformation of Lck, and the phosphatase CD45 preferentially dephosphorylates this tyrosine residue. Once Y505 has lost its phosphate group, Lck can be autophosphorylated in trans, leading to its enzymatic activation. Interestingly, CD45 can play a dual role in the TCR/CD3 signaling cascade, as, together with its capacity to remove phosphate groups from CD3 ITAMs, it is also able to dephosphorylate Y394 (and other tyrosine residues), turning off Lck activity [42]. Indeed, in the absence of the CD45 phosphatase, Y505 of Lck is hyperphosphorylated, which stabilizes the closed conformation and prevents the trans-autophosphorylation of the activation loop [43,44,45]. Once activated, Lck phosphorylates its substrates, the ITAM sequences in the cytoplasmic tails of CD3 heterodimers and ζ-chains, which are located in close proximity, and the phosphorylated ITAMs serve as docking sites to recruit and activate the kinase ZAP70, which, in turn, phosphorylates the tyrosine residues of its substrates (see below).

## 4. ZAP70 Is Recruited and Activated after ITAMs Phosphorylation

ZAP70 was first described in 1991 by the Arthur Weiss group as a tyrosine-phosphorylated protein that binds to the ζ-chain after TCR stimulation [46]. ZAP70 belongs to the Syk family of protein tyrosine kinases (PTKs), which includes only Syk and ZAP70. Syk and ZAP70 show a 56% overall sequence identity, and both PTKs show different expression patterns, with Syk being expressed by B cells, macrophages, monocytes and platelets, and ZAP70 showing a narrower expression pattern, mainly restricted to T cells and NK cells. Shortly after the first description of ZAP70, the same group was able to clone the corresponding cDNA, demonstrating that this molecule was a novel PTK [47]. Interestingly, this paper showed that the binding to the ζ-chain and the tyrosine phosphorylation of ZAP70 required the presence of Src-family PTKs, providing, for the first time, a mechanism by which the Src-family PTKs and ZAP70 may interact to mediate TCR signal transduction. The ZAP70 structure comprises two tandem SH2 domains and a carboxy-terminal kinase domain. The SH2 domains are separated by a linker region, termed interdomain A, and they were shown to specifically bind to the tyrosine-phosphorylated ζ-chain and CD3ε polypeptide [48]. A second unstructured region, referred to as interdomain B, connects the SH2 domains to the kinase domain (Figure 4A). This region contains three tyrosine residues that are phosphorylated upon TCR activation and participate in the regulation of ZAP70 enzymatic activity: tyrosines 292, 315 and 319. The amino acid sequence of the kinase domain shows two tyrosine residues in positions 492 and 493 in human ZAP70, which are located in the activation loop of this domain, and the phosphorylation of these tyrosines leads to its enzymatic activation [49] (Figure 4B). The mutation to phenylalanine of tyrosines 492 and 493 of ZAP70 blocked downstream signals, supporting their critical functional role [50]. Therefore, after its Lck-mediated phosphorylation, ZAP70 is sequentially recruited close to the TCR/CD3 complex and enzymatically activated, allowing for the phosphorylation of its substrates, LAT and SLP-76, generating the LAT signalosome [10,51,52].

As for the in vivo relevance of the *Zap70* gene, the opposite occurred with this kinase than with most recently cloned genes, as human immunodeficiencies associated with defects in this gene were first described before a knockout mouse model was generated [53,54,55]. These works demonstrated that mutations affecting the kinase domain or preventing the expression of this PTK generated a Severe Combined Immune Deficiency (SCID) that prevented normal T-cell development, with the absence of peripheral CD8+ T lymphocytes, and abnormal peripheral CD4+ T cells that were refractory to TCR-mediated activation. The phenotype of *ZAP70*-knockout mice (*Zap*^−/−^) recapitulated the main features of human patients, but neither CD4+ nor CD8+ T lymphocytes were present in the *Zap*^−/−^ mice [56]. The molecular mechanism explaining the different consequences on the CD4+ subset in humans and mice lacking ZAP70 remains unclear, but it could be due to the overlapping expression of Syk in thymocytes, which may compensate for the absence of the ZAP70 function and allow for the positive selection of only CD4 SP thymocytes. Notably, NK cells, which also express ZAP70, showed normal numbers and function, which can also be explained by the simultaneous expression of Syk in this cell population.

With respect to the biological significance of ZAP70, it is also of interest to highlight its role as a biological factor in Chronic Lymphoid Leukemia (CLL) [57]. CLL is a lymphoproliferative disease disorder involving the monoclonal expansion of mature B cells, the progression and prognosis of which are highly variable depending on several factors. One of these factors is the expression of ZAP70, which has been shown to correlate with unmutated *IGHV* genes and a more aggressive clinical course. It was suggested that ZAP70 expression, which is restricted to the T lineage in normal healthy cells and not expressed in mature B lymphocytes, is a better predictor of disease progression than *IGHV* rearrangements or CD38 expression [58]. This raised the question of whether ZAP70 has a biological function leading to disease progression, and several reports have shown that the presence of ZAP70 enhances B-cell-receptor (BCR) signaling induced by anti–immunoglobulin M (IgM) stimulation, although tyrosine kinase activity appears to be dispensable for this [59,60,61]. Therefore, these results supported the role as an adaptor molecule for ZAP70 promoting BCR signaling and/or survival in CLL cells. In support of this, it has been recently reported that ZAP70 expression protects CLL cells from spontaneous apoptosis in the absence of BCR engagement, and the reduction in the ZAP70 expression in CLL cells does not affect the intensity of the BCR signaling in their system [62]. In fact, there is evidence, obtained via the use of a mutant of ZAP70 that can be rapidly and specifically inhibited, that this tyrosine kinase has biological functions that depend on its ability to act as an adaptor [63,64]. Therefore, it can be stated that, although its main biological function in healthy T cells is tyrosine phosphorylation, ZAP70 also has other functions as an adaptor molecule, which serve to regulate certain cellular functions, as we will see below.

## 5. Activated ZAP70 Kinase Generates the LAT Signalosome

As already mentioned, once activated, ZAP70 phosphorylates two scaffold proteins, LAT and SLP-76 [51,52,65]. The *Lat* gene was initially cloned in 1998 by the group of Lawrence Samelson after a long search for the gene coding for an intensely tyrosine-phosphorylated protein after TCR-mediated activation, with a molecular weight between 36 and 38 kDa [52]. Until that time, it was known that Lck and ZAP70 operated sequentially upon T-cell activation, and that the signals generated by these enzymes were critical to activate several transcription factors leading to proliferation and differentiation into effector cells. However, there were “missing” links in this process bridging initial tyrosine phosphorylation with more downstream signaling pathways, such as the phosphorylation of PLC-γ1, the generation of Ca^2+^ influx or the activation of MAP kinases. Proteins of 36–38 kDa were very soon described as important substrates for TCR-activated PTKs [66], but it proved very difficult to isolate p36–38, which had been previously shown to interact with Grb2 and SOS to transduce activatory signals in T cells [67]. In their work, Zhang et al. performed the affinity purification of phosphorylated p36 and then amino acid sequencing, which allowed for the cloning of the corresponding cDNA. LAT is an integral membrane protein that presents a short N-terminal extracellular region, a single region of about 20 hydrophobic amino acids that constitutes the transmembrane segment and a long cytoplasmatic region. LAT presents in its cytoplasmic region nine conserved tyrosine residues that, upon phosphorylation, constitute docking sites for several cytosolic proteins, such as Grb2, Gads or PLC-γ1 (Figure 5A). Once recruited close to the plasma membrane, these proteins can themselves be activated via tyrosine phosphorylation and higher concentrations of their substrates can be found in the plasma membrane. This adaptor is expressed in the membranes of thymocytes, peripheral T cells, NK and mast cells, platelets and pre-B cells.

The first functional analyses of LAT in cell lines demonstrated that this adaptor is essential for the generation of calcium fluxes or the activation of the transcription factor NFAT, as LAT-deficient cells showed no deficiency in proximal signaling events, but downstream pathways were not activated in these cells [68]. Moreover, the four carboxy-terminal tyrosine residues of LAT were required for the activation of downstream signaling [69,70]. This kind of analysis established that conserved tyrosines 7, 8 and 9 (171, 191 and 226 in human LAT) share the responsibility of binding to the SH2 domain of Grb2, tyrosine 6 of LAT (132 in human LAT), binds upon its phosphorylation to the C-terminal SH2 domain of PLC-γ1 and the Gads-SLP-76 complex binds to phosphorylated tyrosines 7 and 8 [65,69,70]. It has been proposed that the LAT cytoplasmic tail consists of a long flexible segment constituting a kind of “protein fishing line” encompassing several SH2- and/or PTB-binding motifs [10]. Because of its unstructured nature, the cytoplasmic segment of LAT has a larger capture radius than a compact, folded protein with restricted conformational flexibility. After the TCR-induced phosphorylation of the four C-terminal tyrosines of LAT, the recruitment of Grb2, Gads-SLP-76 and PLC-γ1 occurs and a LAT signalosome is formed, which allows for the activation of downstream pathways (Figure 5B).

The need for the LAT adaptor for T-cell development was formally demonstrated after the generation of a mouse *LAT*-knockout strain (*Lat^−/−^*) [71]. The most remarkable result of LAT deficiency was the total absence of peripheral αβ and γδ T cells due to an early block of thymic development at the CD4−CD8− DN stage. Interestingly, NK cells, which also express LAT, were phenotypically and functionally normal, initially discarding any role for this adaptor in NK-cell biology [71,72]. Next, the in vivo role of the four C-terminal tyrosine residues of LAT was analyzed by means of a knockin strain harboring Tyr-to-Phe mutations at positions 6, 7, 8 and 9 (136, 175, 195 and 235 in mouse LAT, *Lat^4YF^*) [73]. The phenotype of *Lat^4YF^* knockin mice was identical to that of *Lat^−/−^* mice: there was no peripheral T lymphocytes because of a block in T-cell development at the DN stage, but no effect was observed on B cells or NK cells. These results demonstrate that the distal four tyrosine residues of LAT are essential for pre-TCR signaling and T-cell development in vivo, suggesting that the assembly of the LAT signalosome is critical for the β-selection checkpoint at the DN stage. More support for the relevance of the four C-terminal tyrosine residues was obtained by means of adoptive transfer into irradiated *Lat^−/−^* mice with bone-marrow cells (also from *Lat^−/−^* mice) transduced with a retroviral vector coding for wild-type or different LAT mutants [74]. The adoptive transfer of irradiated *Lat^−/−^* mice with bone-marrow cells expressing a LAT mutant containing only the four distal tyrosines (i.e., having the first five tyrosines mutated to phenylalanines) restored thymic development similar to bone-marrow cells expressing wild-type LAT, indicating that the four C-terminal tyrosine residues of LAT were sufficient for T-cell development.

In this context, it was quite surprising that the phenotype of a knockin strain of mice expressing a LAT mutant in which the sixth tyrosine had been mutated to phenylalanine (*Lat^Y136F^*) [75,76]. This tyrosine residue is located in a conserved consensus sequence (YLVV) for the binding of PLC-γ1 [77], and its mutation abrogated the LAT-PLC-γ1 interaction and Ca^2+^ influx generation in Jurkat cells [69,78]. Concordantly, young *Lat^Y136F^* mice showed a partial block in thymic development, with thymi that contained approximately tenfold fewer cells than their wild-type counterparts, and reduced numbers of CD4+CD8+ DP cells. Interestingly, in mice older than 7 weeks, DP cells were almost undetectable, and coincident with the loss of DP cells, a population of CD4 T cells started to dominate the thymus, which corresponded to abnormal peripheral CD4 cells that expanded in the periphery of *Lat^Y136F^* mice. Indeed, spleen and lymph nodes from knockin mice were greatly enlarged and contained a population of CD4+ T cells with a phenotype (CD44^high^, CD62L^low^, CD69^+^ and CD24^−^) resembling activated/memory T cells and spontaneously producing high amounts of T_H_2 type cytokines. These cytokines were responsible for the eosinophilia observed in the thymi and periphery of these mice and the observed augmented population of activated B cells producing high amounts of IgG1 and IgE. Therefore, besides its role as a transducer of activatory signals from the TCR/pre-TCR, LAT showed, for the first time, its role as a regulator of the homeostasis of the T-cell compartment. Shortly thereafter, our group also described how the mutation to phenylalanine of the last three tyrosine residues of LAT (*Lat^Y7/8/9F^* mice) totally blocked αβ T-cell development but allowed γδ T cells to reach peripheral lymphoid organs [79]. Interestingly, this γδ T-cell population also produced T_H_2 type cytokines and gave rise to a lymphoproliferative disorder. Again, this unexpected phenotype suggested that LAT may regulate the signaling cassettes operated by the TCR/pre-TCR complexes.

We will address the question of the negative-feedback loops involving Lck, ZAP70 and LAT later, but it is of interest to note that it was initially unclear whether the effects observed in *Lat^Y136F^* and *Lat^Y7/8/9F^* mice were due to a dysregulated signaling cascade in the periphery or were due to altered CD4 or γδ T-cell maturation. To shed light on this, our group developed a strategy based on conditional knockins and adoptive transfer in CD3-ε-knockout mice, demonstrating that the expression of a *Lat^Y136F^* allele in mature peripheral CD4^+^ T lymphocytes spontaneously generated the same lymphoproliferative disorder involving T_H_2-type cytokines [80]. Furthermore, the elimination of the LAT adaptor in peripheral T cells also generated hyperactivated and hyperproliferative T cells [80,81], supporting that the phenotype observed in LAT-knockin strains was not the result of defective central tolerance but was due to aberrant thymic selection.

It is also of interest to highlight the potential role of LAT in the clustering of the signaling elements of the TCR signalosome. It has been shown that LAT-Grb2-Sos1 interaction is able to generate liquid condensates at the plasma membrane, which are capable of triggering TCR signaling due to their ability to concentrate signaling reactants, as well as to exclude CD45 [82,83]. Indeed, it has been very recently shown that LAT condensates are thermodynamically coupled to ordered membrane domains during T-cell activation, demonstrating that coupling of membrane domains and cytoplasmic condensates via LAT is essential for activating signaling downstream of TCR ligation [84].

## 6. Regulation of Lck Activity

As previously discussed, Lck is the first kinase to act upon TCR engagement. The phosphorylation of residues Y394 and Y505 correlates with the different conformations of Lck, which have been classically related to its activity. CD45 is crucial for Y505 dephosphorylation and the opening of the Lck conformation. The CD45 phosphatase possesses a large extracellular domain that is highly glycosylated, which is required for optimal TCR signaling [85]. Interestingly, in this work, Acuto and coworkers also showed that Lck is phosphorylated at Y394 in unstimulated Jurkat cells. This study challenged the model, in which it was TCR engagement that served to dephosphorylate Y505 and consequently open the Lck structure. More support for a model in which the opening of the Lck structure was not the only requisite for TCR signaling came from FRET analysis, which did not detect any conformational change in Lck after TCR ligation [86]. Shortly thereafter, the same group demonstrated that relatively high amounts of Lck kinase phosphorylated at Y394 are expressed in resting T cells and thymocytes [87]. In their work, they also showed that approximately half of the Lck molecules phosphorylated at Y394 were also phosphorylated at the inhibitory Y505 residue, constituting what was observed, for the first time, as a doubly phosphorylated Lck (DPho-Lck). Interestingly, this Dpho-Lck isoform showed the same kinase activity as Lck phosphorylated only at Y394. Moreover, in their work, Nika et al. showed that TCR stimulation did not increase the activation of Lck (measured as phosphorylation at Y394), contrary to other downstream signals. Given that the enzymatic inhibition of Lck prevented TCR signaling, these data suggested that a pool of Lck molecules are basally preactivated to phosphorylate ITAMs as soon as they are located in the proximity, but if the Y394 residue is dephosphorylated, this would block TCR signaling. Whether the Lck activity upon productive TCR engagement is increased or not is still the subject of debate. Although the maintenance of the pool of Lck phosphorylated at Y394 is necessary for the generation of TCR proximal signals, it has been shown that this active pool of Lck in resting T cells might actually be smaller than originally estimated [88].

This work therefore established that, in resting T cells, there was a pool of Lck molecules ready to act. The question then was whether the pool of activated molecules was fixed or depended on an equilibrium governed by the competing activities of CD45 and Csk kinase (both acting on the Y505 residue of Lck). In a fixed-pool model, the activation of Lck would require changes in its localization for the initiation of the signaling cascade, whereas in the dynamic equilibrium model, a small change in the CD45 or Csk activities would be sufficient to alter the Lck activity and initiate TCR signaling. By means of expressing a mutant form of the Csk (Csk^AS^) kinase that could be inhibited very quickly and specifically, Arthur Weiss’ group showed that the inhibition of Csk^AS^ resulted in potent and sustained signal transduction and cell activation independent of TCR ligation [89]. These data suggested that there is a continuous turnover of the Y394 and Y505 phosphorylation of Lck in resting cells, allowing for the rapid and efficient phosphorylation of ITAMs by Lck in response to TCR stimulation. Later on, the same group generated mice expressing only Csk^AS^ by means of BAC transgenesis into heterozygous Csk^+/−^ mice (as Csk knockout is lethal during embryonic development) [90,91]. The inhibition of Csk^AS^ in thymocytes induced a rapid hyperactivation of Lck, which, in turn, induced the phosphorylation of proximal TCR-signaling components. However, the inhibition of Csk^AS^ did not increase Erk phosphorylation or intracellular Ca^2+^, revealing that the initial activation of Lck was a necessary but not sufficient requirement for the full activation of the TCR-associated-signaling cascade. Moreover, using this murine model, it was shown that Csk inhibition during TCR stimulation generated more potent and prolonged intracellular signals [92]. Interestingly, Csk inhibition enhanced immune responses to very weak antigens, revealing the important role of the negative regulation of Lck by Csk in establishing the signaling threshold and self/non-self-discrimination of the TCR. The model of the interplay between the opposing enzymatic activities of CD45 and Csk is not incompatible with other constraints on the initiation of TCR signaling. For example, it has been shown that the rigid extracellular domain (ECD) of CD45 sterically excludes from the membrane sites of TCR-ligand engagement [93]. It is of interest to note that Csk is a cytosolic protein, and for the negative regulation of Lck activity, it has to be located at the plasma membrane. One of the candidates for recruiting Csk to the plasma membrane is the transmembrane adaptor phosphoprotein associated with glycosphingolipid-enriched microdomains (PAG) [94]. PAG is strongly tyrosine-phosphorylated in unstimulated T cells, allowing it to bind to Csk. Upon TCR stimulation, PAG is dephosphorylated and dissociates from Csk, which would diminish or abolish the suppressive effect of Csk on Lck [94,95]. Moreover, although PAG KO mice do not exhibit obvious alterations in T-cell development or naive T-cell phenotype, effector T cells from these mice show increased activation responses [96]. TCR-stimulated PAG KO effector T cells also show increased phosphorylation of ZAP70 and PLC-γ1 and increased calcium fluxes compared to wild-type T cells, although little or no increase in the phosphorylation of other downstream effectors such as Erk, Akt or JNK.

It therefore seems clear that the balance between the phosphorylated and dephosphorylated states of Y505 and Y394 of Lck, governed in turn by Csk and CD45, establishes an activation threshold for the TCR-signaling cascade. This raises the question, however, of how the recruitment of Csk (a cytosolic enzyme) to the plasma membrane is regulated. Recent evidence has shown that the PD-1 ligation induces PAG phosphorylation and, thus, Csk recruitment [97] (Figure 6). Interestingly, this study also shows a correlation of the PAG expression levels with the increased survival from several tumor types. This study, apart from suggesting that PAG is a critical mediator of PD-1 signaling and thus a potential target for enhancing the anti-tumor action of CAR-T cells, proposes a mechanism by which PD-1, and perhaps other membrane molecules, participate in the negative-feedback loops of the TCR-signaling cascade.

On the other hand, it has been shown that residues Y394 and Y505 are not the only ones that regulate the conformation and/or activity of Lck. In an interesting work, Weiss and colleagues showed an unpredicted role for a tyrosine residue (Y192) within the SH2 domain of Lck [98]. Mutation of this residue strikingly decreases TCR early signaling, including ZAP70, LAT, PLC-γ1 and Erk phosphorylation, and calcium influx generation. These data suggest that Y192 of Lck allows for the interaction with CD45, which in turn is able to dephosphorylate Y505 and stabilize the open conformation of Lck. Therefore, TCR engagement induces Lck-mediated ZAP70 activation, which in turn triggers signaling events, also including negative-feedback loops; one of these loops consists in the phosphorylation of tyrosine 192 of Lck, which disrupts the ability of CD45 to interact with and activate Lck via the dephosphorylation of Y505 (Figure 7). Interestingly, the defects observed in TCR-associated signals produced by mutations in Y192 are strikingly similar to the phenotype originated by CD45 loss, which supports the role of Lck Tyr192 in the interaction with this phosphatase. Accordingly, cells expressing Lck-Y192A or Lck-Y192E mutants show the increased phosphorylation of this inhibitory C-terminal tyrosine residue [98]. This is an unusual regulatory mechanism by which a phosphorylation event triggers the uncoupling of two proteins, and this should be the reason why the mutation to phenylalanine of tyrosine 192 does not have a negative impact on TCR signaling: Phe residues could be structurally not so different from Tyr residues, and are still able to bind to CD45, while Tyr to Ala or Glu induces a strong modification of the tertiary structure of the SH2 domain, preventing binding to CD45 and thus leading to a high basal level of Tyr505 phosphorylation, which sharply reduces the pool of active Lck. Accordingly, knockin mice expressing a Tyr to Glu (Y192E) form of Lck showed a strong decrease in the total thymocyte numbers, with reduced numbers of DP and SP populations and defective positive and negative selection [99,100]. Interestingly, T cells from LckY192E knockin mice showed a diminished binding to CD45 and a concomitant hyperphosphorylation of Y505, thus corroborating previous data obtained from Jurkat T cells. Surprisingly however, KI mice with a Tyr-to-Phe mutation (Y192F), which prevents its phosphorylation by ZAP70, do not appear to show any alteration in thymic development [99]. If Y192 phosphorylation is part of a negative-feedback loop, then an increase in T-cell activation, or alterations in thymic development, would be expected. More work will be needed to clarify these points.

Regarding the functional regulation of Lck, it should be noted that, as a membrane protein, its interaction with membrane lipids affects its localization and functions. Lipids surrounding integral membrane proteins (termed boundary lipids or lipid sheaths) can contribute to the structure and function of these proteins. Two hypotheses exist to explain the interactions between integral membrane proteins. First, the raft hypothesis postulates that, below a critical temperature, there are relatively stable membrane domains of about 100 nm in which specific lipids and proteins assemble. A second model stipulates that more diffuse lipid- and protein-density fluctuations occur in the plasma membrane that are sufficient to promote some molecular encounters, while making others less likely, consequently leading to membrane organization. Although there is evidence to support a general biophysical mechanism for the reorganization of the local order of membranes [101,102,103], recently, Nika and coworkers analyzed the impact of changing the membrane anchor of Lck with the anchors from other membrane proteins, such as LAT, CD4 and palmytoilation-defective CD4 or CD45 [104]. Unexpectedly, only small differences in the ratio of active Lck were observed after changing its membrane anchor, with the exception of the transmembrane segment of CD45, which severely decreased enzymatically active Lck, probably due to the augmented lateral proximity between Lck and CD45, which increased the dephosphorylation of Y394 by endogenous CD45. Collectively, these data suggest that the proximity at the plasma membrane between CD45 and Lck (and thus the activation of the latter) is regulated by the lipids surrounding both proteins. More support for this model comes from molecular dynamics simulations of full-length Lck open and closed conformations using data available from different crystallographic studies, suggesting that Lck interacts with lipids differently in the open and closed Lck conformations, demonstrating that lipid interaction can potentially regulate Lck’s conformation and, in turn, modulate T-cell signaling [105]. This is concordant with the fact that the co-clustering of Lck and the ζ-chain allows for the phosphorylation of ITAMs, even at very high CD45 densities [42]. Progress in this field involves detailed analyses of the boundary lipid composition of Lck and CD45, although this still remains a difficult technical challenge.

With respect to Lck activity regulation, it has to be considered that it binds to CD4 and CD8 co-receptors through an ion-dependent mechanism involving cysteine residues at positions 20 and 23. In spite of the different immunological functions of cytotoxic CD8+ and helper CD4+ T cells, the TCR-signaling pathways are very similar in both cell populations. Although many approaches have been used to analyze the relevance of co-receptors in Lck activity, it remains unclear whether Lck has a functionally equivalent role in both CD4+ and CD8 [106]. Very recently, Horkova et al. generated knockin mice expressing a Lck isoform bearing Cys-to-Ala mutations at positions 20 and 23 (*Lck^CA^*) to address the relevance of the interaction between Lck and CD4/CD8 co-receptors [107]. It was previously shown that Lck-co-receptor interaction is needed for the co-receptor–LCK interactions in the positive selection of T cells [108]. In their recent report, Stepanek and coworkers demonstrate that, although Lck^CA^ molecules are able to transduce pre-TCR and TCR signaling, Lck bound to CD4 and CD8 drives T-cell development and effector immune responses via qualitatively different mechanisms. For example, CD8-bound Lck is not required for antiviral and antitumor activity of cytotoxic T cells in mice, although it enables CD8+ T cell responses to suboptimal antigens. By contrast, the binding of Lck to CD4 is needed for the proper development and function of helper T cells. Very interestingly, differences were observed in the response to high- and low-affinity antigens. The differential role of co-receptor-bound Lck in the response to high- and low-affinity antigens sheds new light on the regulation of TCR signaling and should facilitate the rational design of T-cell-based immunotherapies.

## 7. Keeping ZAP70 in Check

Lck activity leads to ITAMs phosphorylation in CD3 and the ζ-chain, and doubly phosphorylated ITAMs recruit ZAP70 kinase through its tandem SH2 domains. The binding to phosphorylated ITAMs triggers the opening of the ZAP70 conformation, allowing the phosphorylation of various tyrosine residues. As already mentioned, phosphorylation at Y492 and Y493 stimulates its kinase activity, whereas that at residues Y292, Y315 and Y319 serve to regulate its signaling functions. Indeed, Tyr-to-Phe mutation of residue 319 (*ZAP70^Y319F^*) dramatically impairs LAT and SLP-76 tyrosine phosphorylation, NF-AT activation and IL-2 production [109]. Shortly thereafter, the same group demonstrated that Y319 in the interdomain B of ZAP70 mediates the SH2-dependent interaction between Lck and ZAP70 [110]. Tyr to Phe mutation at this residue impeded ZAP70-Lck interaction, giving rise to a molecular explanation for the block in the intracellular signaling of the *ZAP70^Y319F^* mutation. Strikingly, the amino acid sequence encompassing Y319 (YDSP) interacted with the Lck-SH2 domain with a lower affinity than an alternative phosphopeptide containing the sequence YEEI. In their report, Acuto and his colleagues expressed in Jurkat cells a gain-of-function mutant of ZAP70 by changing the sequence Y319SDP into Y319EEI, which caused a prominent increase in TCR-associated intracellular signals, even though its enzymatic activity and binding capacity to Lck were similar to the normal form of ZAP70, unveiling that the Y319-mediated binding to the SH2 domain of Lck is crucial for the propagation of the signaling cascade leading to T-cell activation. Concordantly, in a transgenic mouse model in which the ZAP70 gain-of-function (ZAP-YEEI) mutant was expressed, CD8 T cells showed that ZAP-YEEI expression produced an increase in basal LAT and Erk phosphorylation, as well as at short times (30 s) after stimulation with anti-CD3 antibody [111]. However, at longer times TCR-dependent intracellular signals and IFN-γ production were decreased in CD8 T cells from Tg-ZAP-YEEI mice, suggesting that other downstream negative-feedback mechanisms were being activated in an increased manner. Together with experiments performed with P116 ZAP70-deficient Jurkat cells [112], these data suggest a model in which the Lck-mediated phosphorylation of Y315 and Y319, in addition to the recruitment of downstream activatory and inhibitory effector molecules, also relieves an autoinhibitory intrinsic interaction, allowing for its kinase activity.

Consistent with the important role of tyrosines in the interdomain B of ZAP70, mutation of some of these residues to phenylalanine drastically affected thymic development [113,114]. Indeed, Y315F and Y319F mutations attenuated positive and negative selection [113,114], while the Y292F mutation upregulated proximal TCR-signaling events. The negative regulatory role played by Y292 residue is probably mediated via its ability to recruit the Cbl and Cbl-b ubiquitin–protein ligases. Along this line, the ZAP-Y292F mutation diminished the internalization and degradation of the TCR/CD3 complex in response to antigenic stimulation, and Cbl recruitment to the immune synapse was also retarded in ZAP-Y292F T cells [115]. Similarly, the mutation of both Y315 and Y319 tyrosines via alanine residues (YYAA) in mice impaired T-cell development and, strikingly, favored the development of rheumatoid factor antibodies, but failed to develop autoimmune arthritis [116]. The crystal structure of ZAP70 confirmed the relevance of tyrosine residues in the interdomain B, showing that Y315 participates in a hydrophobic interaction with W131 in interdomain A, and Y319 interacts with the N-lobe of the catalytic domain [49,117,118]. Accordingly, Trp-to-Ala mutation (W131A mutation) results in increased responses to TCR stimulation in Jurkat cells [117]. Although *ZAP70^W131A^* knockin mice exhibited relatively normal T-cell development, the crossing to OT II TCR transgenic mice unveiled an increase in the negative selection of OT II+ thymocytes, and also in the numbers of T regulatory cells [119]. Interestingly, *ZAP70^W131A^* mice also showed increased expression of several anergy-related genes.

It is of interest to note that the kinase activity of ZAP70 triggers negative regulatory loops. Taking advantage of a ZAP70 mutant (ZAP70-AS) that can be efficiently and specifically inhibited [120], Arthur Weiss and coworkers used mass spectrometry and phosphoproteomics to analyze the differential pattern of phosphosites in response to ZAP70 inhibition [121]. In their report, they show that, besides the expected reduction in the phosphorylation of downstream substrates, ZAP70 kinase activity inhibition induces increased phosphorylation of ITAMs and Y394 of Lck, and this inhibitory loop seems to be mediated by the phosphorylation of Y192 in Lck (see above). As we have already mentioned, this potential negative regulatory loop is inconsistent with the lack of effect of the Tyr-to-Phe mutation at Y192 in Lck, and more experimental work should shed light on its biological relevance.

Other mechanisms have been shown to regulate ZAP70 activity. It has been shown that PKCθ directly interacts with Y127 of ZAP70, and that the interdomain residues Y315 and Y319 are also needed for PKCθ-ZAP70 binding [122]. Interestingly, introducing mutations in PKCθ that block phospho-tyrosine binding prevents not only PKCθ-dependent signaling events, such as nuclear factor κB (NF-κB) activation, but also the phosphorylation of LAT or PLC-γ1, signaling proteins that are traditionally considered to be activated independently of PKC. In addition, it has been proposed that cysteine 564, located in the kinase domain of ZAP70, is palmitoylated, and that a Cys-to-Arg mutation preventing palmitoylation blocks TCR signaling and T-cell activation [123]. However, it has been recently shown that, although such a mutation prevents palmitoylation, a ZAP70 mutant in which cysteine 564 is replaced by a non-palmitoylable alanine residue is able to transduce TCR signaling, and even enhances the activity of Lck and increases its proximity to the TCR [124]. Therefore, C564 seems to be another relevant component in the regulation of proximal TCR signaling by ZAP70 kinase. The generation of animal models with this mutation will certainly be helpful to better understand the biological implications of this regulatory node.

## 8. Negative-Feedback Loops Involving LAT

Different approaches for the analysis of TCR early events have shown a transitory interaction between the open active form Lck and the LAT adaptor that could downregulate the kinase activity of Lck, and this interaction seems to require a conserved stretch of negatively charged amino acids between residue 113 and residue 126 of LAT [125,126]. We have shown that the substitution of this segment with an uncharged segment has a dual role transmitting TCR incoming signals, increasing proximal signaling events, but leading to a reduction in more downstream signaling events, such as Ca^2+^ influx generation or Erk phosphorylation [127]. In 2019, the Arthur Weiss group shed more light on this point, proposing a model in which Lck plays a role as an adaptor molecule indirectly bridging ZAP70 with its substrate LAT [128]. In their report, Lo et al. show that Lck associates with a conserved proline-rich motif (PIPRSP) in LAT via its SH3 domain, and with phospho-ZAP70 via its SH2 domain, thereby acting as a molecular bridge that facilitates the colocalization of ZAP70 and LAT (Figure 8). Indeed, the elimination of this proline-rich motif (LAT-AIRSA mutant) totally abrogates LAT phosphorylation and other downstream TCR signals. Although a KI mouse model in which these LAT prolines are mutated has not been generated, authors have performed the analysis of thymic development using LAT-deficient hematopoietic stem cells lentivirally transduced to express wild-type LAT or the LAT-AIRSA mutant, showing that this mutation causes a partial block in T-cell development. Therefore, these data demonstrate that Lck is a key player not only in the recruitment and activation of ZAP70, but also in allowing the location of the “active” form of ZAP70 in the proximities of the LAT adaptor through temporally limited interactions, making LAT–Lck interaction a possible mechanism to avoid the prolonged transmission of activatory signals.

In the course of immune responses, antigen-specific T cells are activated and first undergo a phase of clonal expansion, followed by a contraction phase that restores normal T-cell numbers and requires apoptosis. Although Fas engagement is essential for T-lymphocyte apoptosis and immune-system homoeostasis, it has been demonstrated that the simultaneous engagement of TCR, CD28 and Fas in naive human T lymphocytes led to decreased activation and proliferation [129]. In this context, we have described that the LAT adaptor undergoes proteolytic cleavage after Fas or prolonged CD3 stimulation [130]. This proteolytic cleavage generates N-terminal truncated forms of LAT that lack the essential C-terminal tyrosines but still retain the proline-rich region and the negatively charged segment, and theoretically would still be capable of Lck binding. The proteolytic cleavage of other adaptors with intracellular-signaling functions has also been demonstrated [131,132,133]; thus, this may be a general mechanism of the termination of intracellular signals coupled to immune receptors. In this regard, a family has been described in which mutations in LAT resulting in the generation of a premature stop-codon and protein truncation leads to a novel form of severe combined immunodeficiency [134]. This mutation generated a form of LAT truncation shortly after the transmembrane domain, and the affected individuals had extremely low numbers of CD4 and CD8 T cells, consistent with the impaired T-cell development observed in LAT-deficient murine models [71]. Another study in humans describes a homozygous mutation in exon 5, leading to a premature stop codon deleting most of the cytoplasmic tail of LAT, and presenting a more complicated phenotype with some unexpected clinical features [135]. In their work, Keller et al. show that three patients homozygous for this mutation, which generates a truncated form of LAT lacking the critical C-terminal tyrosine residues for signal propagation, presented with combined immunodeficiency and severe autoimmune disease from early childhood. Contrasting the *Lat^4YF^* knockin mice (in which the last four C-terminal tyrosines were mutated to phenylalanine residues) [73], reduced numbers of T cells were present in the patients, and they were able to induce Ca^2+^ influx and nuclear factor (NF) kappa B signaling, unveiling unpredicted functions for the N-terminal region of LAT preceding the four C-terminal tyrosine residues, which could play a regulatory role to be analyzed in more detail.

Another point of the negative regulation of TCR signaling appears to be localized to the sixth tyrosine of LAT (human number 132, mouse number 136). This residue has the exclusive ability to recruit PLC-γ1, which is crucial for Ca^2+^ mobilization, Erk and PKC activation, and eventually the irreversible activation of T cells [136]. However, phosphorylation kinetics of this residue is slow compared to other phosphorylation sites on LAT [137]. The slow phosphorylation kinetics of this relevant tyrosine residue is due to the presence of an evolutionarily conserved glycine just preceding the tyrosine, which makes it a poor substrate for ZAP70, as the kinase domain of ZAP70 strongly prefers an acidic residue (aspartate or glutamate) at the −1 position relative to the substrate tyrosine residues [138]. This slow kinetics suggested a molecular mechanism to support the kinetic-proofreading (KPR) model of TCR ligand discrimination [139] (Figure 9). This model proposes that, after TCR-pMHC binding, there is a time lag between TCR engagement and irreversible T-cell activation, and, consequently, the longer the pMHC-TCR interaction, the better the T-lymphocyte activation. In 2019, Lo et al. examined the effects of replacing the glycine residue at position 131 (G131) in the human form of LAT with aspartate (LAT-G131D) or glutamate (LAT-G131E), and found that this leads to a remarkable and specific increase in LAT-Y132 phosphorylation, PLC-γ1 phosphorylation and Ca^2+^ influx generation [140]. Moreover, they found that the difference between wild-type LAT and the LAT-G131D or LAT-G131E mutant increase is remarkably higher when cells are stimulated with low-affinity peptides, supportive of the proposal of an important role for the phosphorylation kinetics of the sixth tyrosine residue of LAT in the discrimination between self and non-self. A recent report, in which mass spectrometry-based methods were used to quantify the first molecular events triggered after TCR stimulation with peptides of varying affinities has lent support to this model [141]. In their work, Roncagalli and coworkers show that differences in the signaling between strong ligands and weak ligands was not reflected by changes in very early signaling events, but rather by changes in later events focused around ZAP70 activation and the phosphorylation of the signaling adaptor LAT.

Our group also contributed to give support to this model, as well as to unveil the potential of accelerating Ca^2+^ influx generation to regulate peripheral tolerance [142]. Besides verifying that a LAT^G131D^ mutant increased Y132 phosphorylation and Ca^2+^ fluxes in Jurkat cells, we showed that cells expressing the LAT^G131D^ mutant secrete greater amounts of interleukin-2 (IL-2) in response to CD3/CD28 engagement, but cells expressing the LAT^G131D^ mutant were more sensitive to the inhibition of IL-2 production via pre-treatment with anti-CD3, which points to a possible role of this residue in the generation of anergy. We and others have generated a knockin strain of mice in which the glycine residue preceding the sixth tyrosine residue (Y136 in mice) was substituted by an aspartate (*Lat^G135D^* mice) [143,144]. Consistent with the in vitro results, T cells from *Lat^G135D^* mice showed increased peripheral T-cell activation and proliferation, a larger number of anergic T cells, fewer peripheral CD8 T cells and more γδ-T cells than their wild-type littermates. Remarkably, the LAT^G135D^ mutation is dominant, as heterozygous mice also showed an altered phenotype compared to wild-type mice [143]. Also of interest is the fact that *Lat^G135D^* mice showed enhanced self-reactivity, which increased thymic negative selection [144]. Moreover, Lo et al. performed *Listeria* infection experiments, showing that T cells from *Lat^G135D^* mice proliferate more than their wild-type counterparts in response to very weak stimuli, but display an imbalance between effector and memory responses. Strikingly, furthermore, despite increased negative selection and anergy, mice expressing the LAT^G135D^ mutant exhibit features associated with autoimmunity and immunopathology. Altogether, these results show that the phosphorylation of LAT at Y136 constitutes an important step for the kinetic-proofreading model, which requires the proper conditions, as both signaling deficiency and hyperactivity can lead to immunodeficiency and/or autoimmunity. This duality of the LAT adaptor provides another indication of the level of complexity and subtlety in controlling the development of immune responses.

## 9. Concluding Remarks

Tyrosine phosphorylation is a key post-translational modification that constitutes an essential mechanism of signal transduction in eukaryotic cells. In the immune system, this process is widely represented, and in T cells, the first biochemical event that occurs after antigenic recognition seems to be the phosphorylation of the tyrosine residues included in the ITAM sequences of the CD3 chains. For T cells to develop, differentiate and proliferate, proper intracellular signaling is required, and the negative regulation of such signaling pathways is crucial to prevent autoimmune diseases or severe immunodeficiencies. Although not currently understood in depth, it is becoming increasingly clear that multiple negative regulatory loops exist along the TCR-signaling cascade. The tyrosine kinases Lck and ZAP70, together with the transmembrane adaptor LAT, are essential players in the transduction of TCR early signals, and the elimination of any of them causes serious alterations in the TCR-signaling cascade. During the last 25 years, the generation of new relevant data on TCR signaling has been constant, but it is in the last decade that we have begun to understand the importance of the regulation of early TCR signals.

These new data offer a more detailed view of the first molecular events that occur after the TCR binding to antigens of varying affinities, generating slightly different signalosomes, and show how a delicate control over tyrosine phosphorylation via substrate selection and spatial localization fills in long-standing gaps in the kinetic-proofreading model. The new knowledge also raises new questions to be solved. In this review, we have mainly focused on three early players, which, besides their ability to transduce positive/activation signals, have intrinsic autoregulatory functions allowing for balanced responses during T-cell development and activation. The description of motifs controlling the speed of tyrosine phosphorylation in the LAT adaptor, the modification of which increases the antigenic sensitivity of the TCR, together with a better understanding of the interactions of Lck, ZAP70 and LAT, helps us to better understand relevant biological processes, and broadens the range of new therapeutic approaches for the treatment of immune-based diseases, or the design of CAR-T cells for the treatment of cancer. The possibility of generating base-edited T cells in which some of the amino acids that regulate the regulatory functions of these molecules are changed could be a promising approach to trigger the therapeutic potential of CAR-T cells.

## Figures and Tables

**Figure 1 biology-12-01163-f001:**
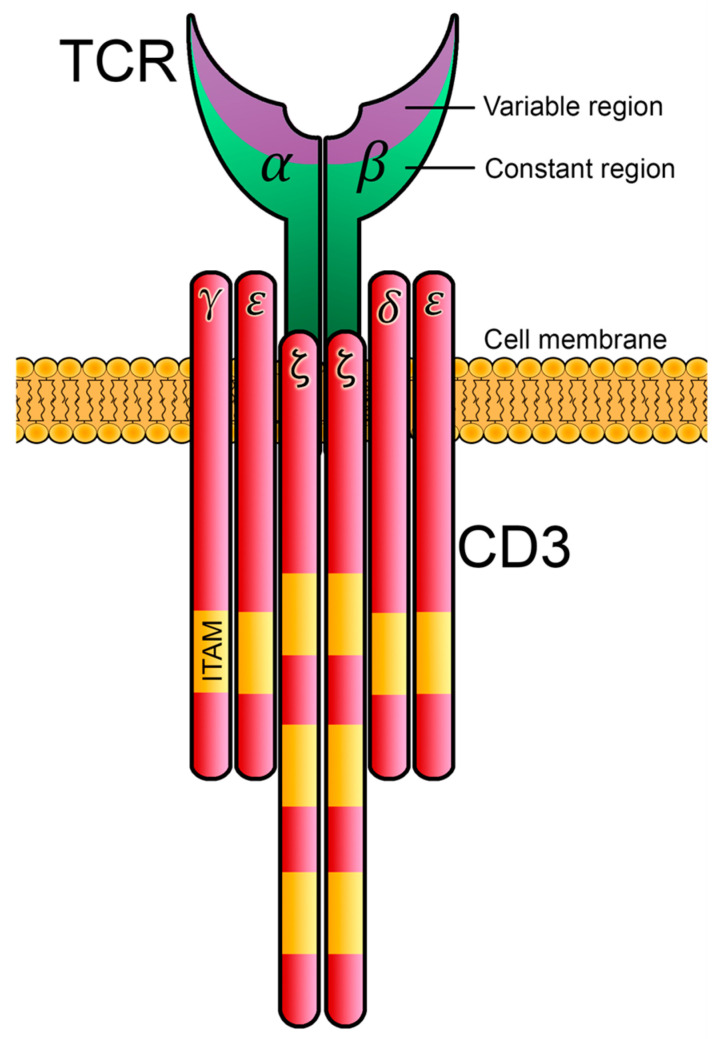
**Structure of the TCRαβ/CD3 complex**. The T-cell receptor is constituted by heterodimers of α and β chains. The TCRα and TCRβ polypeptides have variable and constant immunoglobulin domains, and both chains are bound by covalent disulfide bonds. TCRαβ heterodimers and CD3 polypeptides are associated via non-covalent bonds. CD3 is constituted by CD3ε, CD3γ, CD3δ and ζ-chain polypeptides, organized as dimers (CD3εγ, CD3εδ and CD3ζζ). ITAMs are located in their intracellular tails, allowing for the transmission of information from the TCR inside the cell.

**Figure 2 biology-12-01163-f002:**
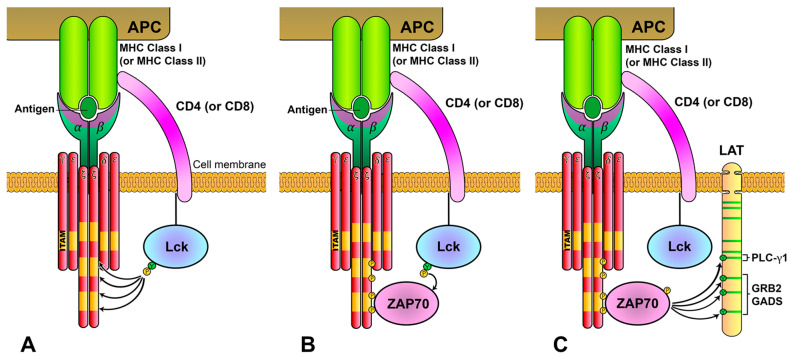
**TCR/CD3-complex-mediated intracellular signaling.** (**A**) TCR/CD3 complex recognizes antigens bound to MHC-I or MHC-II presented by an antigen-presenting cell (APC), which allows the activity of Lck kinase over the ITAMs located in CD3 ζ-chains. (**B**) Phosphorylation of ITAMs generates docking sites for ZAP70 kinase in the proximities of active Lck, which, in turn, phosphorylates tyrosines of ZAP70, activating its kinase domain. (**C**) Catalytically active ZAP70 preferentially phosphorylates tyrosines in the cytoplasmic tail of the LAT adaptor, generating binding sites for SH2 domains, leading to the formation of a signalosome.

**Figure 3 biology-12-01163-f003:**
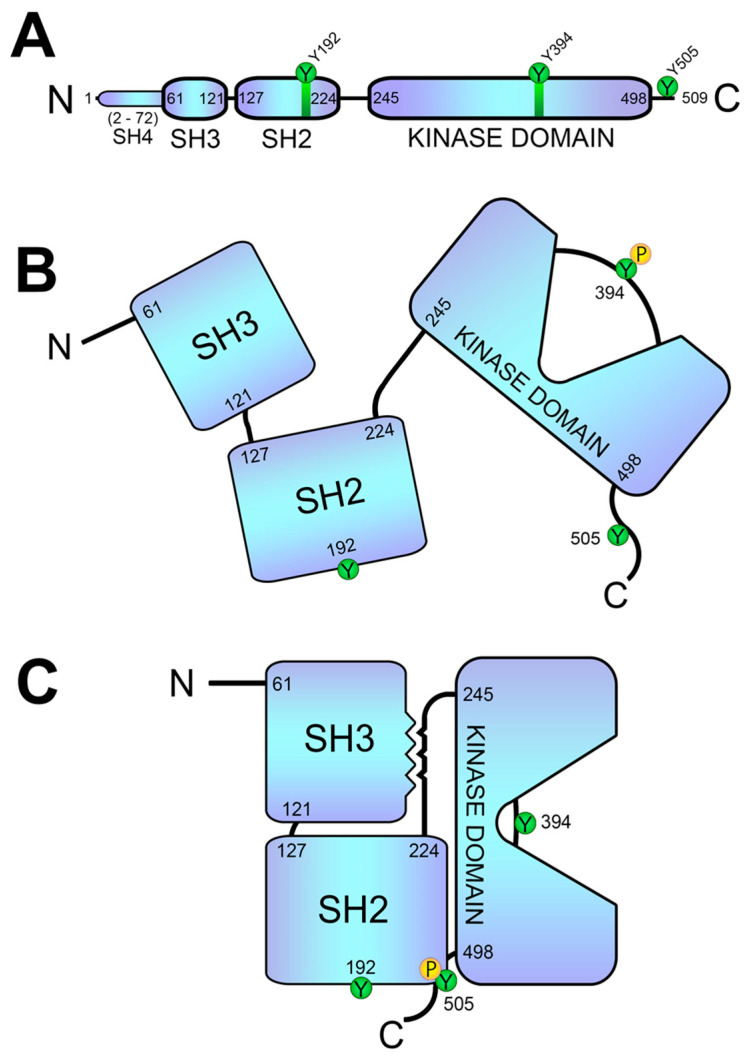
**Structure and conformations of the Lck kinase.** (**A**) Linear sequence of Lck, showing its SH4, SH3, SH2 and catalytic domains. (**B**) Representation of the unfolded (open) Lck conformation. Tyrosine 505 (Y505) regulates conformational changes required for Lck activation. (**C**) Representation of the folded (closed) conformation of Lck. The phosphorylation of Y505 generates an SH2-binding site, causing a folded conformation in which the Y394 is not exposed. This closed conformation is further stabilized via the interaction of the SH3 domain with the proline motif located between the SH2 and kinase domains.

**Figure 4 biology-12-01163-f004:**
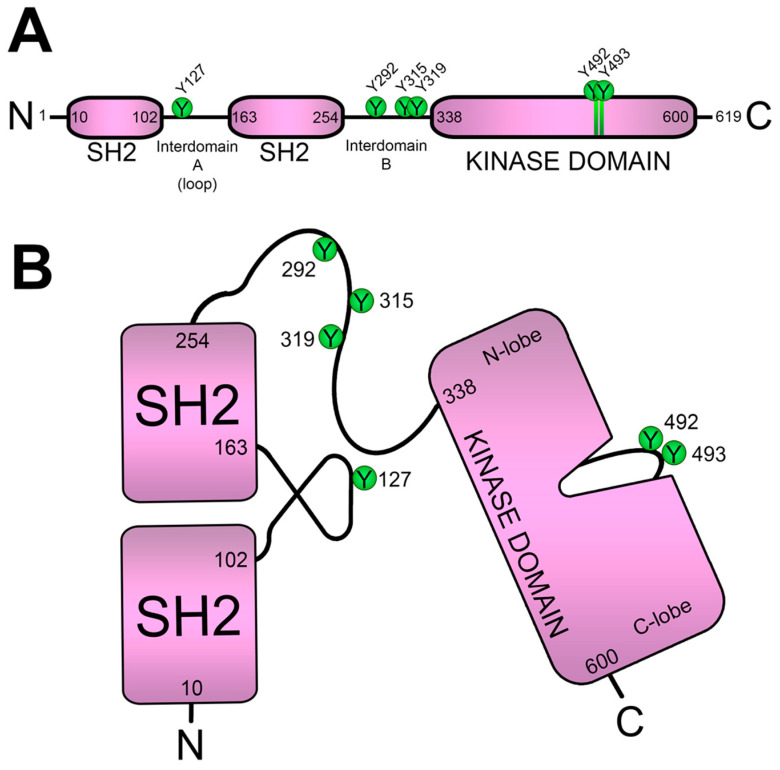
**Structure of the ZAP70 kinase.** (**A**) Linear sequence of ZAP70, showing its two tandem SH2 domains linked by interdomain A, and interdomain B that connects the second SH2 domain with the C-terminal kinase domain. (**B**) Representation of the ZAP70 conformation. The two SH2 domains bind specifically phosphorylated tyrosine residues in the CD3 ITAMs. Tyrosines 292, 315 and 319, responsible for ZAP70 enzymatic activity regulation, are located in interdomain B.

**Figure 5 biology-12-01163-f005:**
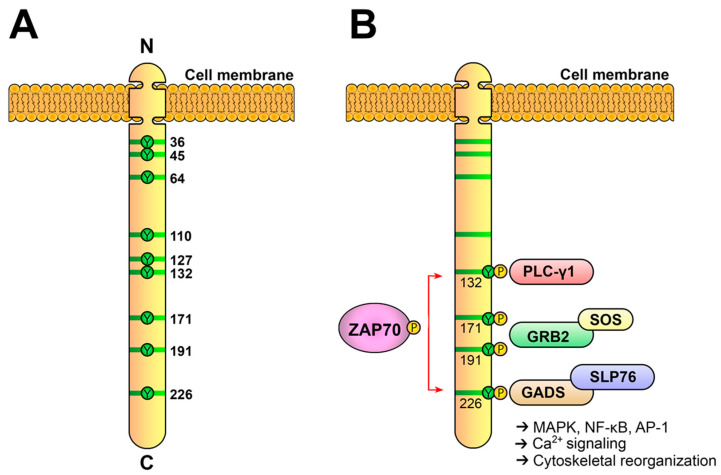
**The LAT adaptor acts as a signaling scaffold.** (**A**) Schematic representation of human LAT. LAT is an integral membrane adaptor with a short N-terminal extracellular region, a transmembrane region and a long cytoplasmic tail. This tail contains nine conserved tyrosine residues that are phosphorylated to give rise to new binding sites for the SH2 domains of proteins such as PLC-γ1, Grb2 and Gads. (**B**) LAT signalosome formation. The LAT signalosome is a multiprotein complex formed by LAT, PLC-γ1, Grb2, Gads and SLP-76.

**Figure 6 biology-12-01163-f006:**
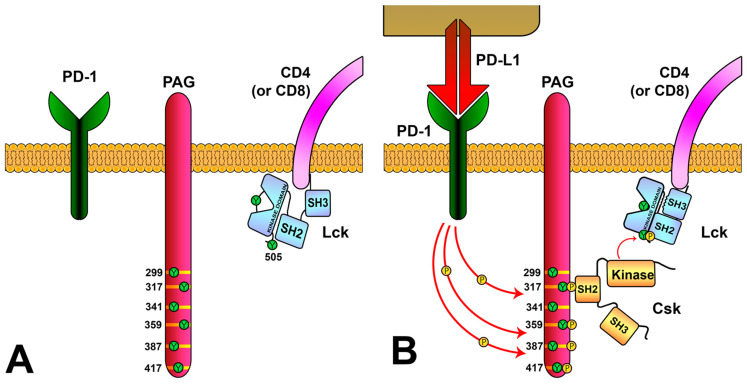
**Regulation of Lck activity via the PD-1/PAG pathway.** (**A**) PAG is located in the T-cell membrane and has multiple tyrosine residues in their cytoplasmic region. (**B**) PD-L1 binding to PD-1 induces the phosphorylation of PAG tyrosines, generating binding sites for Csk. The SH2 domain of Csk binds to PAG-phosphorylated tyrosine 317, causing a conformational change that increases its kinase activity. Csk localizes close to the membrane and phosphorylates tyrosine 505 of Lck, inducing Lck to its inactive conformation.

**Figure 7 biology-12-01163-f007:**
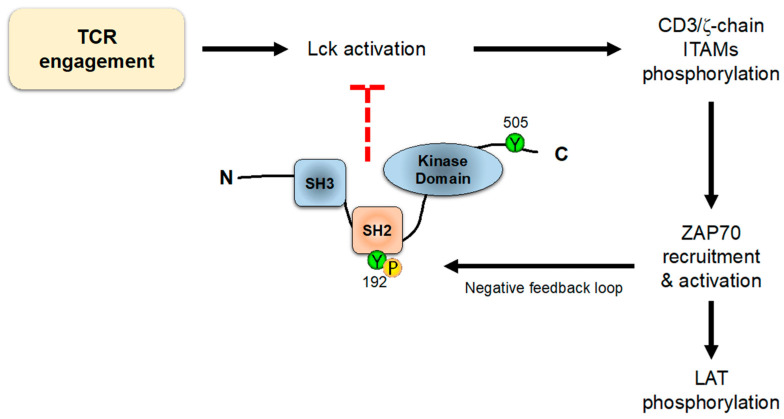
**Lck negative-feedback loop.** TCR engagement activates intracellular signaling leading to ITAMs phosphorylation and ZAP70 recruitment. The activation of ZAP70 leads not only to downstream activation signaling, but also to the phosphorylation of Y192 of Lck, which inhibits its binding to CD45 phosphatase, allowing the phosphorylation of Y505 and promoting the closed Lck conformation.

**Figure 8 biology-12-01163-f008:**
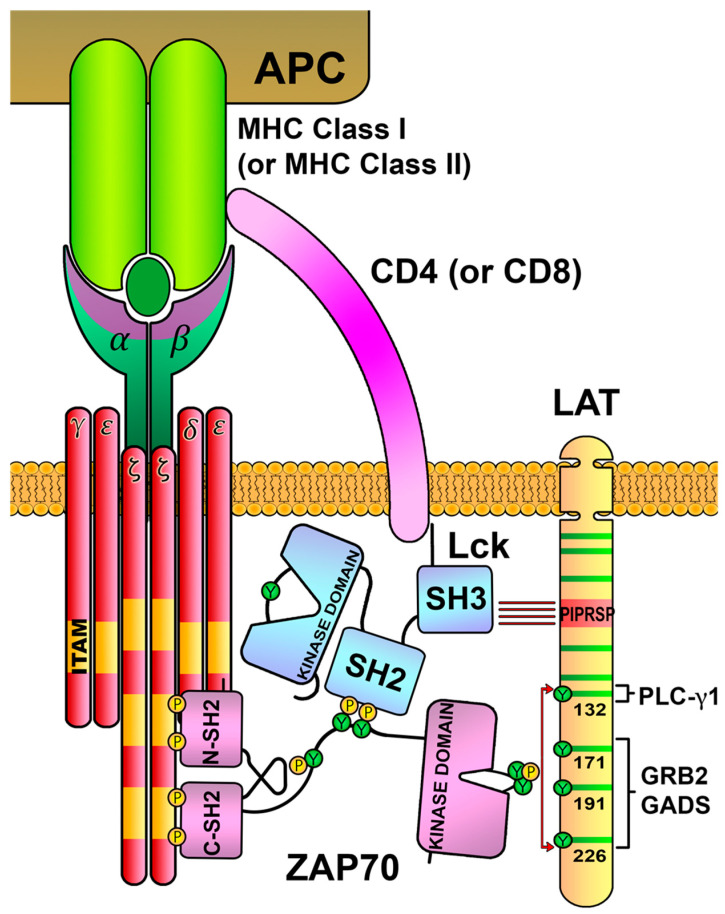
**Interactions connecting Lck, ZAP70 and LAT.** The SH2 domain of Lck binds to the phosphorylated tyrosines in the interdomain B of ZAP70, and the SH3 domain of Lck binds to the proline-rich region of LAT (PIPRSP). This binding bridge generated by Lck localizes the ZAP70 kinase close to the LAT adaptor, favoring its tyrosine phosphorylation.

**Figure 9 biology-12-01163-f009:**
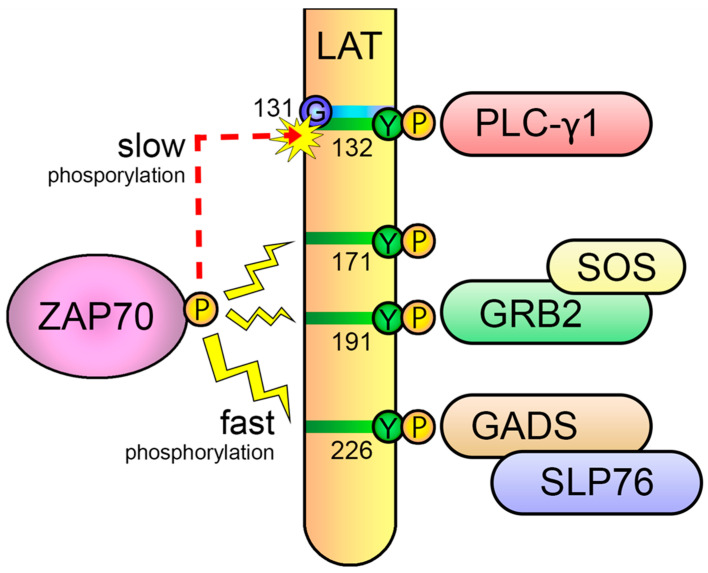
**Slow phosphorylation of the sixth tyrosine of LAT regulates TCR signaling.** The evolutionarily conserved glycine residue preceding the sixth tyrosine of LAT (Y132 in human LAT) makes it a worse substrate for ZAP70, leading to a slower phosphorylation of this residue with regard to the other C-terminal tyrosines. This slow kinetics phosphorylation is a molecular mechanism that supports the kinetic-proofreading (KPR) model of TCR ligand discrimination.

## Data Availability

Not applicable.

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
