# Peer review of "A Story of Kinases and Adaptors: The Role of Lck, ZAP-70 and LAT in Switch Panel Governing T-Cell Development and Activation"

_biology, 2023, doi:10.3390/biology12091163_

Round 1

Reviewer 1 Report

The review is clear and give a very thorough overview of the role of kinases in TCR signalling and T cell activation. 

I have only one request regarding the scientific content, which is related to the notion of aggregation, or clustering of TCR, Lck, and the scaffolding proteins that build up TCR signalosome. If there is some controversy about TCR or Lck clustering, the notion that Lat, Grb2 and SOS form aggregates (sometimes referred to as condensates or even phase separation) to amplify signalling is not debated, as far as I know. The review must include a paragraph, even a short one (or critical), about this notion.

Otherwise, I have only a few comments and suggestions. 

Lines 61-62: the sentence implies that TCR signalling only, skew the differentiation of T cell into different subsets. This is not the case, as cytokine signalling is considered as the key input pushing T cells into one or another subset. The sentence must be rewritten to avoid this confusion. 

Figure 2. Regulation of Zap70 by Lck is discussed much later in the review and not on page 4, which is related to figure 2. Nevertheless, the figure implies that Lck does not phosphorylate Zap70, which is confusing and must be corrected. In the text, a short mention of this phosphorylation event could help, but that is not required (this is a good illustration of issues that can arise from discussing the same topic in several different sections of the review). 

Line 118: At the surface (instead of “in”) 

Lines 125-141: This entire paragraph is almost fully redundant with the text starting line 70. Some sentences are almost the same. For the sake of ease of reading, it must be removed or merged with the text above. 

Lines 169-182: This paragraph could be removed, as the review is fairly long already and its content is not crucial (it was once thought that Nck plays a role, but now it is accepted it does not). 

Lines 222-225: There is a great summary of how the various models involving conformational change during TCR triggering in this review, http://dx.doi.org/10.1038/nri2887, maybe it could help to cite it here. 

Lines 289-290: It may be relevant here to mention that CD45 also very heavily dephosphorylate ITAMs on TCR ( https://doi.org/10.1038/nsmb.2762

Lines 361-364: Although Art Weiss is already cited many times in this review, it might suitable here to mention this paper in the context of Zap70 functioning as an adaptor  ( https://doi.org/10.5281/zenodo.6113017 ). 

Lines 613-614: It would be relevant here to mention that co-clustering of Lck and CD3zeta allows ITAM phosphorylation of ITAMs even at very high CD45 densities (https://doi.org/10.1038/nsmb.2762 , fig. 7) 

Author Response

The review is clear and give a very thorough overview of the role of kinases in TCR signalling and T cell activation.

We thank the reviewer for her/his comment. We have tried to make the review as complete as possible and there is always the question of whether relevant work has been omitted. We do appreciate an independent reviewer who tells us that the work is very thorough.

I have only one request regarding the scientific content, which is related to the notion of aggregation, or clustering of TCR, Lck, and the scaffolding proteins that build up TCR signalosome. If there is some controversy about TCR or Lck clustering, the notion that Lat, Grb2 and SOS form aggregates (sometimes referred to as condensates or even phase separation) to amplify signalling is not debated, as far as I know. The review must include a paragraph, even a short one (or critical), about this notion.

We appreciate the suggestion. In the new version of the manuscript we have included a small paragraph explaining the relevance of Lat, Grb2 and SOS aggregates in TCR signaling (lines 451-458).

Otherwise, I have only a few comments and suggestions.

 Lines 61-62: the sentence implies that TCR signalling only, skew the differentiation of T cell into different subsets. This is not the case, as cytokine signalling is considered as the key input pushing T cells into one or another subset. The sentence must be rewritten to avoid this confusion.

Thank you very much for the comment. It is certainly true that, as it was written, this sentence could be confusing with regard to the differentiation of T lymphocytes. We have included a short sentence (line 62) explaining the relevance of cytokines, without going into more detail because it was beyond the scope of this review.

Figure 2. Regulation of Zap70 by Lck is discussed much later in the review and not on page 4, which is related to figure 2. Nevertheless, the figure implies that Lck does not phosphorylate Zap70, which is confusing and must be corrected. In the text, a short mention of this phosphorylation event could help, but that is not required (this is a good illustration of issues that can arise from discussing the same topic in several different sections of the review).

We are grateful to the reviewer for bringing this point to our attention. It is true that the previous version of Figure 2 could be misleading. It has been corrected to show that Lck phosphorylates ZAP70 once it has been recruited to ITAMs.

Line 118: At the surface (instead of “in”)

Thank you for the correction of this typo (now on line 124).

Lines 125-141: This entire paragraph is almost fully redundant with the text starting line 70. Some sentences are almost the same. For the sake of ease of reading, it must be removed or merged with the text above.

This paragraph has been deleted to make the article easier to read. It certainly was, for the most part, redundant.

Lines 169-182: This paragraph could be removed, as the review is fairly long already and its content is not crucial (it was once thought that Nck plays a role, but now it is accepted it does not).

Thank you again for the comment. The paragraph has been almost completely removed, leaving only one sentence to make the text understandable below (see lines 157-159).

Lines 222-225: There is a great summary of how the various models involving conformational change during TCR triggering in this review, http://dx.doi.org/10.1038/nri2887, maybe it could help to cite it here.

We thank the reviewer for her/his help. The reference to the review by van der Merwe and Dushek (# 24) has been included in line 174.

Lines 289-290: It may be relevant here to mention that CD45 also very heavily dephosphorylate ITAMs on TCR ( https://doi.org/10.1038/nsmb.2762).

The new version of our manuscript mentions this fact, and includes the corresponding reference (# 42, lines 268-270).

Lines 361-364: Although Art Weiss is already cited many times in this review, it might suitable here to mention this paper in the context of Zap70 functioning as an adaptor  ( https://doi.org/10.5281/zenodo.6113017 ).

Unfortunately, in the article mentioned by the reviewer we have not found any data explaining the functions of ZAP70 as an adaptor. The only thing that is mentioned in that article concerning ZAP70 is the fact that blocking transferrin uptake selectively impairs the phosphorylation of Tyr319. However, the comment of the reviewer was appropriate and we have included a sentence explaining very briefly the functions of ZAP70 beyond its kinase activity, with the corresponding references (lines 341-343).

Lines 613-614: It would be relevant here to mention that co-clustering of Lck and CD3zeta allows ITAM phosphorylation of ITAMs even at very high CD45 densities (https://doi.org/10.1038/nsmb.2762 , fig. 7)

A short sentence has been included to address this relevant point (lines 612-613).

Reviewer 2 Report

The article entitled ‘A story of kinases and adaptors: role of Lck, ZAP-70 and LAT 2 in the switch panel governing T cell development and activation’’ was well received. This review article entails a comprehensive description of T cell activation process especially the role of ZAP70, and Lck.

The idea is interesting, and the authors have demonstrated a deeper understanding of the topic. The paper is well written, containing beautiful and comprehendible illustrations where necessary. The flow of the ideas is fluent and reader friendly.

In concluding remarks the authors have mentioned and I quote ‘The description of motifs controlling the speed of tyrosine phosphorylation in the LAT adaptor, and whose modification increases the antigenic sensitivity of the TCR, together with a better understanding of the interactions of Lck, ZAP70 and LAT, helps us to better understand relevant biological processes, and broadens the range of new therapeutic approaches for the treatment of immune-based diseases, or the design of CAR-T cells for the treatment of cancer’’. I would love to see a separate section about how the current knowledge can open the gates for novel CAR-T designs and its implications in immunotherapy.

Language is fine. No major issues

Author Response

The article entitled ‘A story of kinases and adaptors: role of Lck, ZAP-70 and LAT 2 in the switch panel governing T cell development and activation’’ was well received. This review article entails a comprehensive description of T cell activation process especially the role of ZAP70, and Lck.

We truly appreciate the work and kind comments of the reviewer. In writing this review we tried not to leave out any work of relevance, and it is appreciated that an independent expert view confirms that this is work worthy of publication.

The idea is interesting, and the authors have demonstrated a deeper understanding of the topic. The paper is well written, containing beautiful and comprehendible illustrations where necessary. The flow of the ideas is fluent and reader friendly.

We are grateful for these positive comments.

In concluding remarks the authors have mentioned and I quote ‘The description of motifs controlling the speed of tyrosine phosphorylation in the LAT adaptor, and whose modification increases the antigenic sensitivity of the TCR, together with a better understanding of the interactions of Lck, ZAP70 and LAT, helps us to better understand relevant biological processes, and broadens the range of new therapeutic approaches for the treatment of immune-based diseases, or the design of CAR-T cells for the treatment of cancer’’. I would love to see a separate section about how the current knowledge can open the gates for novel CAR-T designs and its implications in immunotherapy.

We agree with the reviewer that a section on new CAR-T designs would be of great interest to the scientific community. However, our manuscript is already much longer than what we initially intended to write. Therefore, although a supplementary section on new CAR-T designs and their implications in immunotherapy would be of undoubted interest, it is beyond the scope of our review. The statement in the Concluding Remarks section has been expanded a bit to reflect where these potential lines of research might go.
